# An Aero-Engine Damage Detection Method with Low-Energy Consumption Based on Multi-Layer Contrastive Learning

**Xing Huang [1], Lei Li [2,*] , Jingsheng Zhang [3] , Dengfeng Yin [1], Xinjian Hu [2] and Peibing Du [4]**

1 AECC Hunan Aviation Powerplant Research Institute, Zhuzhou 412000, China; caifeichao@nwpu.edu.cn (X.H.); capi@608.163.net (D.Y.)
2 College of Information Science and Engineering, Hunan University, Changsha 410000, China; huxinjian@hnu.edu.cn
3 Eberly College of Science, The Pennsylvania State University, University Park, PA 16801, USA; jxz5511@psu.edu
4 Northwest Institute of Nuclear Technology, Xi'an 710024, China; dupeibing1@nint.ac.cn
* Correspondence: aleilei@hnu.edu.cn; Tel.: +86-182-2996-0127

**Abstract:** The health of aero-engines is pivotal to the safe operation of aircraft. With increasing service time, the internal components of the engine will be damaged by threats from different sources, so it is necessary to regularly detect the damage inside the engine. At present, most of the detection methods of major airlines rely on the internal images of the engine obtained by manual use of a borescope to detect damage or traditional machine learning methods, which consume high levels of human and computational resources but have low efficiency. Artificial intelligence in various fields can achieve better performance than traditional methods, but to achieve the industrialization standard of Green AI, we need further research. Accordingly, we introduce a multi-layer contrastive learning method to a lightweight target detection model design, which is applied to real aero-engine borescope images of complex components to accomplish real-time damage detection. We intensively conduct comparative experiments to evaluate the effectiveness of our method. The verification results demonstrate that the method can help our model perform excellently compared with other available baseline models.

**Keywords:** damage detection; contrastive learning; aero-engine surface damage

## 1. Introduction

Aviation safety is crucial to ensuring people's livelihood, economic development, and military security. As a result of the long period service of an aircraft, it can experience various types of damage to the engine, such as blade damage caused by friction, discoloration caused by high-temperature burning, etc. The engine's complex and compact component structure poses a significant challenge to damage detection.

As the most intuitive carrier of surface information, images are widely used in damage detection and localization. Major airlines usually use a borescope to collect internal images of the engine, then locate and diagnose the damage through manual observation. However, with the rapid development of the aviation industry in recent years and the increasingly urgent need for low emissions, methods that rely on manual detection and traditional machine learning cannot meet the growing demand for detection. At the same time, related technologies based on object detection have been continuously innovated. Object detection constitutes the technical basis for computer vision tasks, such as instance segmentation [1], image captioning [2], and object tracking [3]. Good results have been achieved in many fields, such as face verification [4], pedestrian detection [5], and license plate detection [6].

Therefore, we propose a lightweight damage detection method based on computer vision to complete the task of detecting the damage of aero-engine complex components according to the images collected by the borescope. Compared with the baseline model in the field of damage detection, the proposed method has lower computing power consumption. Our work reduces the burden on technicians and increases the probability of damage

detection and also eliminates the hidden dangers of aircraft to a greater extent, which can be significant.

In summary, the specific work in this paper is as follows:

- We manually classify and label the damaged area of the internal damage images of the engine to form a dataset suitable for supervised learning, focus on the problems of limited borescope image samples and difficult damage labeling, and adopt a relatively new image augmentation method to pre-process images.
- We proposed a multi-layer contrastive learning method to pre-train the "backbone + neck" network of the damage detection model. Positive and negative samples are constructed based on the multi-layer sample features output by the target detection network, and then the pre-training process is completed through self-supervised learning. Finally, we use supervised learning to achieve the downstream task of target detection and fine-tune the parameters of the damage detection network. The method we propose reduces the damage detection algorithm's dependence on the number of labeled samples and improves the damage detection accuracy of the model.
- We conduct extensive experiments with the baseline model to evaluate the ability of damage detection and to verify that the use of the multi-layer contrastive learning to pre-train the "backbone + neck" network can effectively improve the accuracy of damage detection.

## 2. Related Work

### 2.1. Target Detection

Object detection generally includes two sub-tasks: one is to obtain the position of the target in the image, and the other is to obtain the category of the target. Girshick et al. [7] proposed the Recurrent Convolutional Neural Network (RCNN) network, which introduced the convolutional neural network into the target detection task, and obtained a very large performance improvement on the general dataset VOC at that time. He et al. proposed Spatial Pyramid Pooling network (SPPNet) [8], which used SPP operation to bypass the step of adjusting the image size, and improved the detection speed by nearly 20 times on the basis of RCNN. Girshick et al. [9] proposed the Fast RCNN, which integrated the characteristics of SPPNet and RCNN and further optimized it. The detection accuracy and speed were again considerably improved, but it was still unable to reach real-time detection. In the same year, Ren et al. [10] proposed the Faster RCNN detection network, which uses the RPN network to generate high-quality region proposals, which greatly improved the detection speed and initially achieved the effect of real-time detection. The above work divided the detection process into two stages: first obtaining the detection area, then using the convolutional network to return the target position.

However, some researchers have tried to utilize a one-stage network to complete all steps to speed up detection and improve calculation efficiency. In 2015, Redmon et al. [11] proposed the YOLO detection network, which can reach up to 155FPS, but has lower detection accuracy. In recent years, many researchers have developed multiple versions based on YOLO, which have achieved good detection accuracy while maintaining high speed. Liu et al. [12] proposed an SSD network, which used different layers of the network to detect targets of different sizes, greatly improving the detection accuracy. To a certain extent, it alleviated the barriers of a one-stage target detection algorithm in small-sized targets, reducing some computational costs and the negative environmental impact of the AI approach.

### 2.2. Contrastive Learning

MoCo [13] is a momentum-based contrastive learning method proposed by Kaiming He et al. It provides a simple and effective training method for image processing and has achieved very good results in downstream image classification tasks. MoCo adopts the instance discrimination task [14] as an auxiliary task to shorten the distance of the same image in different situations. Chen et al. introduced the image augmentation method

and added a nonlinear transformation between representation and contrastive loss into a network structure, proposing a new simple framework for contrastive learning of visual representations—SimCLR [15]. Due to the collapse problem of self-supervised training, Grill et al. constructed a BYOL network [16] by increasing a predictor network and a stop-gradient strategy. It redistributes the results and eigenvalues, pulls the distance between positive and negative samples, and avoids training degradation. Xie et al. [17] proposed a pixel dimension comparison algorithm based on full supervision. On the basis of the cross-entropy loss, the pixel contrast loss is added. The cross-entropy loss is used for the calculation of the pixel category loss, and the contrast loss improves the pixel feature space by calculating the similarity of the pixels. It can decrease the distance between the pixels belonging to the same category and increase the distance between pixels of different categories while model coding.

### 2.3. Aircraft Engine Damage Detection

At present, the aero-engines internal damage detection methods based on computer vision widely adopt related methods in the field of target detection. They take different damages of components as targets and achieve good results in terms of detection accuracy and speed. Abhishek et al. [18] used CNN to perform loss detection analysis on the collected engine images. Svensen et al. [19] used deep neural networks to classify parts from borescope images of large turbofan engines. Li [20] used the YOLO v3 [21] network for damage detection of the compressor components in the engine. He used DenseNet [22] to integrate some high-layer features into the backbone network, Darknet53, which enhanced the ability of feature propagation, feature reuse, and feature fusion. These deep-learning-based aero-engine damage detection methods still remain in the theoretical stage and have not been put into practical application. At the same time, due to the complexity of the industrial scene, it is difficult to obtain borescope images, and it takes much manpower to label the damaged area. Thus, there are insufficient samples for supervised training. The above reasons cause high resource consumption, low detection accuracy, and poor generalization of the detection model.

In this paper, we combine target detection and a multi-layer contrastive learning method for pre-training. With images obtained by the endoscopic techniques used as the information source, our goal is to label the location, contour, and confidence of the damage on the image in real-time and accurately in a more green way.

## 3. Methodology

In this section, we will introduce our proposed method in detail. Our method consists of two parts, which include a pre-training architecture design based on self-supervised learning and the design of a surface damage detection network for aero-engine internal components based on the YOLOX model.

### 3.1. YOLOX Target Detection Network

The YOLO series model has iterated five versions since the first version was released by Jeseph in 2015. Under the premise of ensuring low energy consumption of the model, YOLOX takes the YOLO v3 model as the basic framework and adds the technologies Decoupled Head, Anchor Free, and SimOTA to solve the problems of the original network. The structure of the YOLOX target detection model, shown in Figure 1, can be divided into three parts: the backbone network (CSPDarknet53), the neck network (feature pyramid), and the detection head.

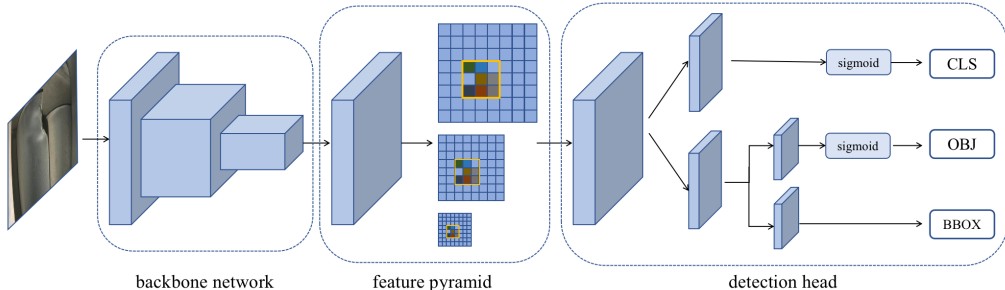

**Figure 1.** The structure of the YOLOX target detection model.

### 3.1.1. Backbone Network

The backbone network follows the CSPDarknet53 network, which improved upon the Darknet53 network in YOLO v4 [23]. The network is a fully convolutional network (FCN) that does not contain fully connected layers, mainly composed of convolution blocks and CSP modules. On the network, to avoid using the pooling layer to lose many detailed features, only downsampling is used to shrink the image. The low-layer features are very important for the localization of the target.

The convolution block consists of a convolution layer, a batch normalization (BN) layer, and an activation layer. The BN layer can force the data to be redistributed in the region where the activation function is sensitive, which helps the network to run faster and converge to the global optimal point. When the convolutional structure is stacked into deep layers, residual learning [24] is introduced here to eliminate the gradient problem. The activation function used by the activation layer is the SiLU function, and the formula is:

$$f(\mathbf{x}) = \mathbf{x} * sigmoid(\beta \mathbf{x}) \tag{1}$$

where $\mathbf{x}$ is the input feature map and $\beta$ is the hyperparameter. Because it is continuous and differentiable, it enables the model to be trained easily.

The CSP module divides the input features into two branches for forward propagation. Performing convolution operations in each branch to halve the number of channels, then only performing Bottleneck * N operation on one of the branches, and concatenating the output structure of the two branches so that the input and output of Bottleneck are the same size, allows the model to learn more features.

### 3.1.2. Neck Network

The neck network follows the structure of the feature pyramid network (FPN) [25], which feeds back the top-feature-layer that contains rich semantic information by layer after forward propagation and concatenates to the bottom feature layer that contains rich location information. Different layers lead to detecting branches for detecting objects of different sizes. Based on the FPN structure, PAFPN introduces the path aggregation (PA) idea from the PANet [26]. On the basis of the top-to-bottom feature path possessed by FPN, PAFPN further aggregates the high-layer semantic information and the bottom-layer spatial location information. A bottom-to-top enhancement path is added, shortening the information path and strengthening the capability of the feature pyramid.

### 3.1.3. Detection Head

The YOLOX model adopts the anchor free method, that is, multiple anchor frames of different scales are not preset for each grid point, and each grid point is only responsible for the prediction of one frame. Because the PAFPN structure supplements the spatial information of the feature and the focal loss predicts the target center area, the anchor free method exceeds the anchor based method in detection accuracy and greatly reduces the amount of parameters during training. It also speeds up the inferring of model results.

### 3.2. Pre-Trained Model Based on Self-Supervised Learning

In this section, we propose a multi-layer contrastive learning method for unsupervised visual representation learning based on momentum contrast (MoCo) to train the CSPDark-Net53 + PAFPN as an encoder without labels by mining the supervision information of the damaged image samples themselves. We construct the auxiliary task of "checking dictionary" to improve the coding similarity between anchor samples and positive samples, reduce the coding similarity with negative samples, and finally enable the encoder to complete the encoding at the abstract semantic feature space.

The structure of the training method framework is shown in Figure 2.

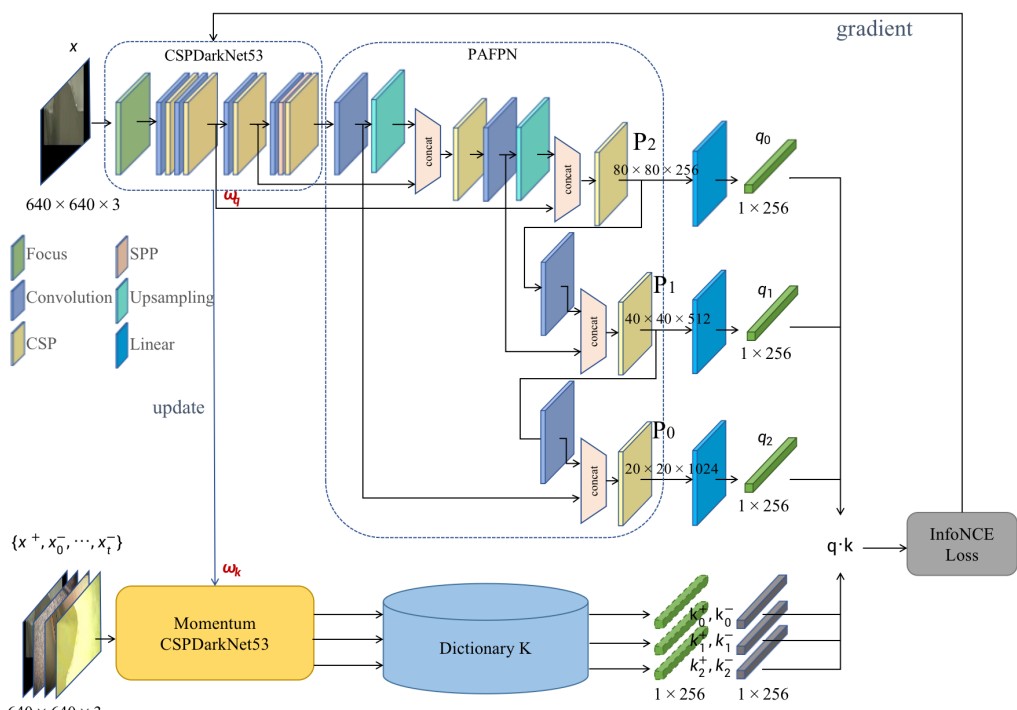

**Figure 2.** The encoder uses five cascaded CSP modules to downsample the features and then uses an SPP module to increase the receptive field of the network. The feature map is subjected to three maximum pooling of different sizes through this module. The feature pyramid fuses the output features of the backbone network, aggregates the information of different layers, and then outputs the anchor sample (sample image to be trained) features through linear transformation. The anchor sample features are compared with the positive and negative samples found in the dictionary, and the feature extraction network is updated to complete the pre-training.

### 3.2.1. Momentum-Based Contrastive Learning

Contrastive learning can be thought of as training an encoder for dictionary lookup. The core idea of MoCo is to maintain a dictionary as a queue of data samples.

In mini-batch training, the magnitude needs to be strictly controlled each time the parameters of the dictionary encoder are updated. First, we randomly select a batch of pictures for two image augmentations and then use two encoders to obtain the sum, use the sum to obtain the similarity of the positive samples, use the sum dictionary queue to obtain the similarity of the negative samples, and finally calculate the contrast loss value. At the same time, we place the current mini-batch into the dictionary queue and remove the earlier mini-batch from the dictionary queue, which can ensure that the encoding in the dictionary is relatively consistent with the prediction dataset in the near future. When the parameters are updated by backpropagation, only the parameters of the target detection model (anchor sample encoder) are updated. Due to the structure of the dictionary encoder

being the same as that of the anchor sample encoder, the parameters are fused in proportion to control the encoding; thus, consistency with the dictionary is guaranteed.

3.2.2. Pre-Training Framework Based on Multi-Layer Contrastive Learning

We obtained the augmented anchor image of the image to be trained by random flipping, cropping, fusion noise, and other image augmentation methods, and then formed a batch of this image and other images randomly selected in the dataset for random image augmentation again, getting the image set $\{\mathbf{x}^+, \mathbf{x}_0^-, \ldots, \mathbf{x}_k^-\}$. The backbone CSPDarkNet53 of the damage detection network proposed in this paper is used as an encoder to encode images, and the structure of PAFPN is used to output three-layer feature maps $\mathbf{P}_0$, $\mathbf{P}_1$, and $\mathbf{P}_2$, layer by layer upward. Feature maps of different sizes correspond to grid divisions of different sizes, which can adaptively balance spatial information and semantic information between detecting targets of different complexity. We add a linear layer to each feature map to transform it into one-dimensional vectors of length 128 as $\mathbf{q}_0$, $\mathbf{q}_1$, $\mathbf{q}_2$. We then use Momentum CSPDarkNet53 to encode the image set $\{\mathbf{x}^+, \mathbf{x}_0^-, \ldots, \mathbf{x}_k^-\}$ in the same way to obtain three-layer one-dimensional vectors of length 128 as $\mathbf{k}_0$, $\mathbf{k}_1$, $\mathbf{k}_2$, and add $\mathbf{k}_0$, $\mathbf{k}_1$, $\mathbf{k}_2$ to the dictionary.

In order to improve the encoder's cognition of damage features under grid divisions of different sizes, we construct multiple pairs of negative samples between the three layers of features output by the feature pyramid so that each feature layer curved out by the current grid size can better represent the damages. After the two encoders encode the same image, the output is the three-layer features. We define the features belonging to the same image extracted by layer in the same depth of encoder and decoder are positive to each other, and features belonging to the layers in different depth or different images are negative to each other.

Taking Figure 3 as an example, after the anchor sample encoder and dictionary encoder encode the same picture, the positive sample is constructed according to $(\mathbf{q}_0, \mathbf{k}_0)$, $(\mathbf{q}_1, \mathbf{k}_1)$, and $(\mathbf{q}_2, \mathbf{k}_2)$. For the task of damage detection, the comparative learning between positive examples and negative examples is introduced to train an encoder that can recognize images from an advanced semantic perspective. Therefore, in order to keep the deeper semantic features of images as much as possible, we increase the proportion of features extracted by the deep network and adopt three sets of negative samples constructed between different layers, namely $(\mathbf{q}_1, \mathbf{k}_0)$, $(\mathbf{q}_2, \mathbf{k}_0)$, and $(\mathbf{q}_2, \mathbf{k}_1)$. At the same time, the anchor sample vector and other feature vectors in the dictionary that do not belong to the picture are all negative samples. We then calculate the contrastive losses of different layers of query in turn and accumulate these contrastive losses as the loss value of a single training. Finally, the network updates the parameters of the encoder CSPDarkNet53 through backpropagation and simultaneously fuses the encoder parameters with the momentum encoder parameters and then assigns the parameters to the momentum encoder. The formula is:

$$\omega_{\mathbf{k}} \leftarrow \sigma\omega_{\mathbf{k}} + (1 - \sigma)\omega_{\mathbf{q}} \tag{2}$$

where $\sigma$ is the constant used to update the parameter weights each time. In order to keep the dictionary consistent, the fluctuation of the model parameter update should be small, so in this paper, $\sigma = 0.999$.

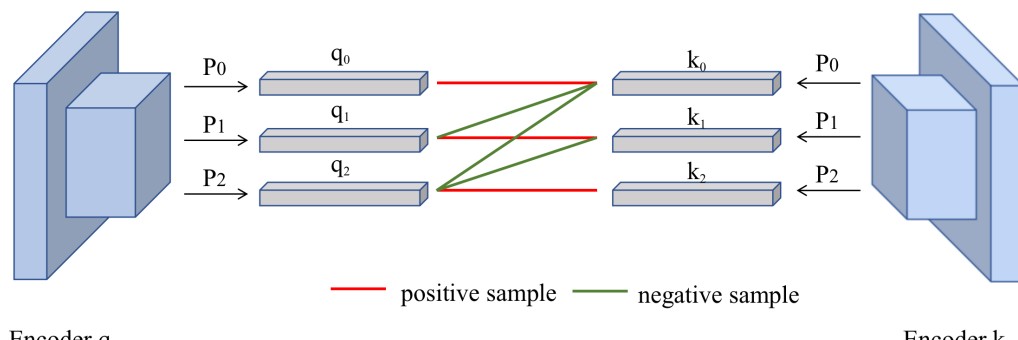

**Figure 3.** Positive and negative sample construction method in the multi-layer.

### 3.2.3. Comparative Loss

We use the three-layer output features of the PAFPN network to perform the training of contrastive learning, and, at the same time, merge the three-layer losses to obtain the overall contrast loss. The formula is:

$$L = -\sum_{i}^{P} \log \frac{exp(\mathbf{q}_i \cdot \mathbf{k}_i^+)/T}{\sum_{j=0}^{K} exp(\mathbf{q}_i \cdot \mathbf{k}_i)/T} \tag{3}$$

where $P$ is the number of layers of the model. The damage detection model used in this paper has three layers, $P = 3$; $K$ is the number of image sets composed of positive samples and negative samples, $\mathbf{q}_i$ represents the query of the $i$th layer, $\mathbf{k}_i^+$ represents the corresponding positive sample, and $\mathbf{k}_j$ represents the contrast sample.

## 4. Dataset

The experimental dataset in this paper comes from the internal damage images of aero-engines obtained by a company using borescope technology in a real environment. From the tens of thousands of raw data, we selected high-pixel images of representative parts with obvious damages, and then classified and labeled these images by component.

### 4.1. Dataset Failure Category

As supervised learning is heavily dependent on the training set data, we screen the damaged images of components, which occupied a large portion of the previous sorting work. Eventually, datasets with three types of aero-engine internal damages for the swirler, the large elbow pipe, and the air compressor/turbine blades were sorted out. An example of the picture is shown in Figure 4.

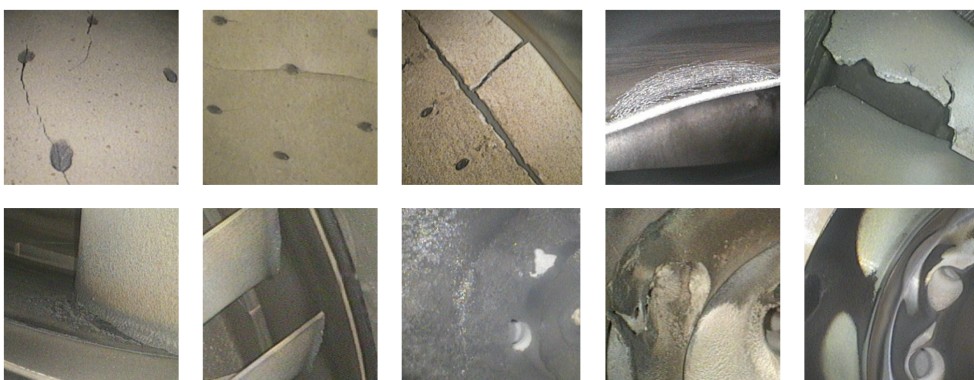

**Figure 4.** Partial damage.

- Swirler: This component is located in the combustion chamber of the engine. The fuel, once injected by the fuel nozzle, is fully mixed with high-speed and high-pressure

air in the swirler. Under ignition, the fuel starts to burn and releases a large amount of heat and pressure. Therefore, the swirler is subjected to a considerable thermal load inside the engine. It includes carbon deposits, metal discoloration caused by high-temperature, and deformation/cracks/loss of components caused by impact. Such damages are fatal to the combustion chamber and seriously threaten the normal operation of the combustion chamber.

- Large Elbow Pipe: This component is located outside the combustion chamber. It has many vents to help exhaust the hot gas after combustion. It is in an extremely high-temperature environment. As a result, its material has problems with cracking and ablation from denaturation caused by high temperature.
- Air Compressor/Turbine Blades: The blades damage dataset was obtained by merging the data of two components, due to the small number of damage images of the compressor blades. For compressor blades, the main threat comes from inhaled foreign matter and oil molecules in the air. Foreign matter will damage or deform the rotating blades, and oil molecules will adhere to the blades, causing engine performance degradation. Turbine blades are near the combustion chamber, which, due to the high-temperature environment, can also cause damage, including ablation/deformation of the blade, blade tip abrasion, and curling caused by friction with the turbine wall.

In this paper, we use *Labelme* to label the damages: outlining the damaged area one dot by one dot, and specifying its classification.

We organize and label three datasets in the paper, which are shown in Tables 1–3.

**Table 1.** Damage statistics in swirler dataset.

|  | **Carbon Deposits** | **Metal Discoloration** | **Loss** | **Ablation** | **Cracks** |
|---|---|---|---|---|---|
| Images | 229 | 63 | 40 | 52 | 26 |
| Label frame | 322 | 93 | 45 | 74 | 40 |

**Table 2.** Damage statistics in large elbow pipe dataset.

|  | **Scratches** | **Ablation** | **Cracks** |
|---|---|---|---|
| Images | 89 | 23 | 113 |
| Label frame | 121 | 36 | 455 |

**Table 3.** Damage statistics in air compressor/turbine blades dataset.

|  | **Loss** | **Tips Whitish** | **Ablation** | **Blade Crimping** | **Abrasion** |
|---|---|---|---|---|---|
| Images | 117 | 83 | 27 | 56 | 37 |
| Label frame | 125 | 219 | 37 | 87 | 39 |

We also counted the damaged area sizes of images in the three datasets, as shown in Figure 5.

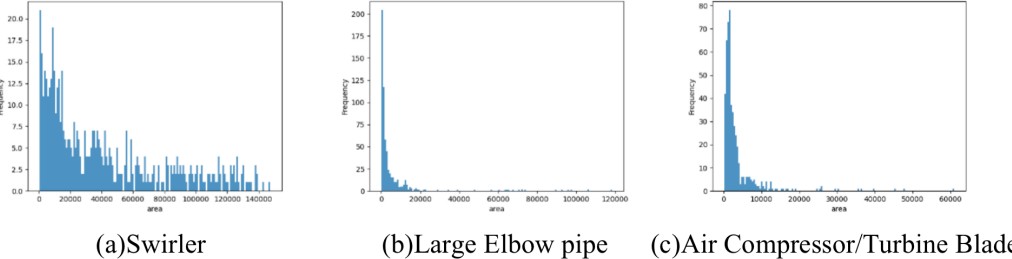

(a)Swirler        (b)Large Elbow pipe     (c)Air Compressor/Turbine Blades

**Figure 5.** The damaged area of the eddy current device is relatively large, the larger part is less than 50,000, and the larger part is more evenly distributed. Most of the damaged areas of the large elbow pipe dataset are less than 10,000, and all of them are below 20,000. Most of the damaged areas of the blade datasets are less than 10,000, but the distribution is smoother than that of the large elbow pipe.

*4.2. Image Augmentation*

For supervised deep learning, enough samples are a prerequisite for training an excellent model. Too few samples will commonly cause the model to fail to learn useful features, or even lead to overfitting.

We adopt the following two image augmentation methods in this paper:

- Mosaic Augmentation: We randomly selected four images from the dataset and used the classical image augmentation method to augment them. We then synthesized the four augmented images into a new image.
- MixUp Augmentation: To augment one image, first, another image from the dataset needed to be selected randomly, and then the two images and their corresponding labels fusion coefficient needed to be fused according to the preset. It can help the model establish a linear understanding of the sample without being disturbed by noise. The formula is:

$$\mathbf{x}' = \lambda \mathbf{x}_i + (1 - \lambda)\mathbf{x}_j \tag{4}$$

where $\lambda \in [0, 1]$, $\mathbf{x}_i$ and $\mathbf{x}_j$ are the input image pixel matrix, and $\mathbf{x}'$ is the output image.

An example of one image after image augmentation shown in Figure 6. After processing the images through the aforementioned two methods: the image samples are added; the detection efficiency of small samples is improved; and the model can learn the subtle differences in the linear samples, thereby expanding the cognitive scope of the model and strengthening the model's ability to analyze outside images of the training set. This can improve the accuracy of the sample prediction and the generalization of the model.

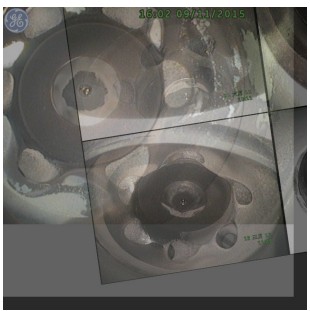

**Figure 6.** An example of the image after Mosaic and MixUp image augmentation.

## 5. Experiments

In this section, we conducted a series of experiments. First, we sifted through the images used to pretrain, train, and test the model. The dataset used for pre-training included 18,313 raw images without labels captured in real scenes, and the dataset of 733 labeled images at a 7:3 ratio, allocated for the training set and test set. The data distributions of the training set and test set are shown in Table 4. To evaluate detection performance, we use average precision (AP) and recall as quantitative measures, and set

intersection over union (IoU) to 0.5:0.95 (i.e., AP50) for detection for the top 100 regions of arbitrary sizes. All the following experimental results are obtained from the test set.

**Table 4.** Dataset division of swirler, large elbow pipe, and air compressor/turbine blades.

|  | **Swirler** | **Large Elbow Pipe** | **Air Compressor/Turbine Blades** |
|---|---|---|---|
| Training Set | 186 | 147 | 176 |
| Test Set | 80 | 65 | 79 |

First, we designed and conducted two sets of contrast experiments. The first set of experiments only used the output feature of the last layer of the CSPDarknet53 + PAFPN for the MoCo method. The output feature tensor size was 20 × 20 × 1024. The pre-trained model was then used as the backbone of the damage detection network to be fine-tuned and trained on the three aero-engine damage datasets. The second set of experiments used all three-layer output features from the CSPDarknet53 + PAFPN to perform pre-training through the multi-layer contrastive learning method proposed in this paper. The pre-trained model also performed tuning training on the three damage datasets. Finally, the gap between the two methods was shown by the AP and recall values of two groups of experiments on the swirler, the large elbow pipe, and the air compressor/turbine blade datasets. The experimental comparison results are shown in Figures 7–9, and a visual comparison example is shown in Figure 10.

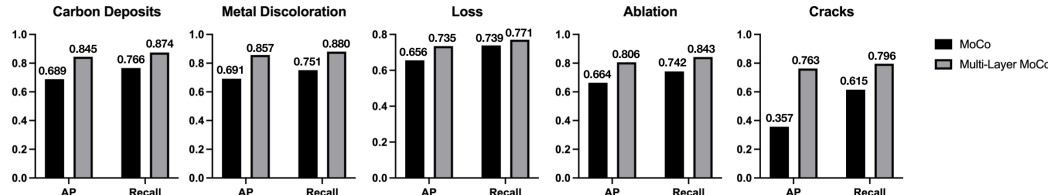

**Figure 7.** Comparison of experimental results between the multi-layer MoCo method and the MoCo method for the swirler dataset.

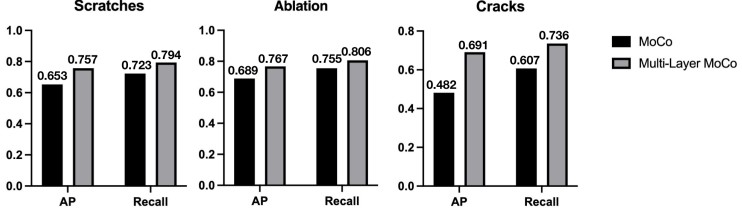

**Figure 8.** Comparison of experimental results between the multi-layer MoCo method and the MoCo method for the large elbow pipe dataset.

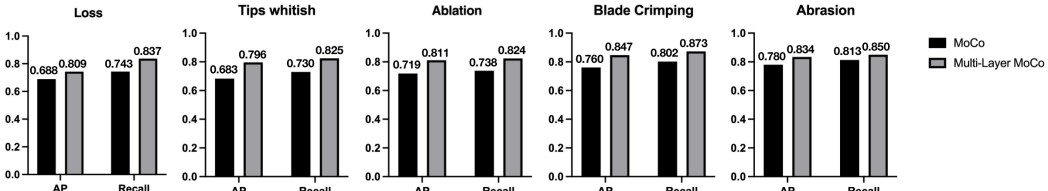

**Figure 9.** Comparison of experimental results between the multi-layer MoCo method and the MoCo method for the air compressor/turbine blades dataset.

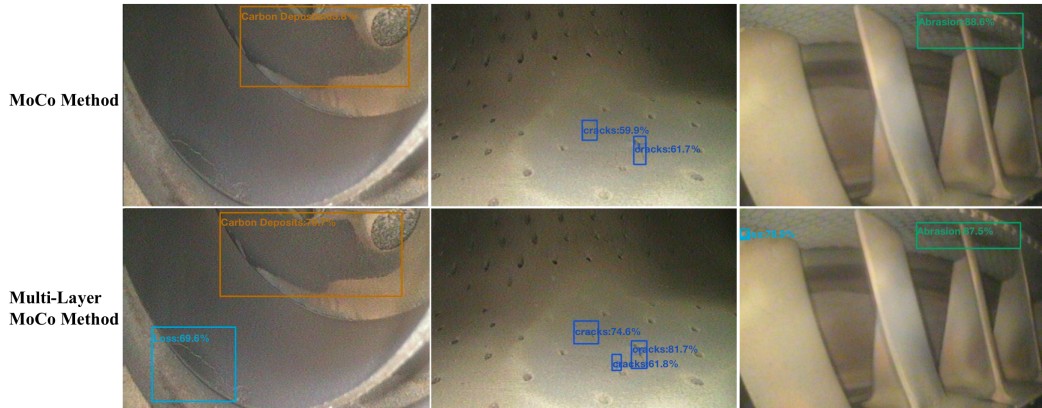

**Figure 10.** Visual comparison example between the multi-layer MoCo method and the MoCo method for the swirler, large elbow pipe, and air compressor/turbine blades. Our detection method is able to detect the damaged areas ignored by the other.

From the results, we see that the use of the multi-layer comparison learning method (multi-layer MoCo) greatly improves the AP value and recall of a single category compared with the MoCo method. Our proposed method benefits from the feature pyramid, which achieves a higher precise representation of different damage, and each layer of its output features is more targeted to provide relatively balanced semantic information and spatial location information. It can also be seen from the table that the AP value of the categories with the lowest detection accuracy using the MoCo pre-training method was greatly improved after using multi-layer comparative learning.

The AP value of crack damage detection in the swirler dataset increased by 20.9%, which indicates that the use of the multi-layer MoCo method can make the output of different layers of the feature pyramid focus on the representation of damage of different complexity, avoiding the problem of data imbalance. The expression of some categories is biased, and finally, the detection accuracy of each category can be improved to a relatively balanced position.

We tested the contribution of the features extracted from each layer in the multi-layer contrastive learning to the model's detection accuracy on the three datasets. Based on the pre-trained model in the above experiments, we attached a detection head to the three feature layers of $80 \times 80 \times 256$, $40 \times 40 \times 512$, and $20 \times 20 \times 1024$ in the CSPDarknet53 + PAFPN to detect the representational capacity of the output features from each layer to all damage on the image. The experimental results obtained are shown in Table 5.

**Table 5.** Performance scores of different feature layers for damage detection in the MoCo method and the multi-layer MoCo method.

| Feature Tensor Size | Dataset | AP | | Recall | |
|---|---|---|---|---|---|
| | | MoCo | Multi-Layer MoCo | MoCo | Multi-Layer MoCo |
| $80 \times 80 \times 256$ | Swirler | 0.066 | 0.144 | 0.119 | 0.157 |
| | Large Elbow Pipe | 0.087 | 0.137 | 0.145 | 0.157 |
| | Air Compressor/Turbine Blades | 0.217 | 0.288 | 0.269 | 0.324 |
| $40 \times 40 \times 512$ | Swirler | 0.330 | 0.409 | 0.379 | 0.438 |
| | Large Elbow Pipe | 0.237 | 0.307 | 0.322 | 0.345 |
| | Air Compressor/Turbine Blades | 0.362 | 0.529 | 0.406 | 0.551 |
| $20 \times 20 \times 1024$ | Swirler | 0.271 | 0.333 | 0.316 | 0.354 |
| | Large Elbow Pipe | 0.403 | 0.468 | 0.448 | 0.495 |
| | Air Compressor/Turbine Blades | 0.406 | 0.333 | 0.427 | 0.348 |

It can be clearly seen from the table that the damage feature extraction ability of each layer substantially increased, verifying that our proposed multi-layer contrastive learning pre-training method can achieve better results on the aero-engine internal component damage datasets. Compared with the baseline model, the detection accuracy of multi-component and multi-category damage of aero-engines is greatly improved.

We wanted to verify that the computational resource consumption of the whole set of damage detection methods proposed in this paper is much lower than that of other baseline models in the industrial application of aero-engine surface damage detection. We used Faster RCNN, DeepLab v3+ with our YOLOX model pre-trained by the multi-layer MoCo method to conduct comparative experiments. We used 509 pieces of images to test the computational performance of each model and finally obtain the result of three quantitative metrics for computational performance of each model—FLOPs, number of parameters, and FPS. FLOPs (floating point operations), also known as the amount of computation, which are usually used to measure the amount of computing resources required by the model. FPS (frames per second) is the number of frames processed per second when detecting video files, which can be used to evaluate the detection speed of the damage detection model in practical application scenarios. The comparative results of computational resource consumption, model size, and inference speed of each model are shown in Table 6.

**Table 6.** Computational resource consumption, model size, and inference speed comparison.

| Model | Image Pixel | FLOPs(G) | Params(M) | FPS |
|---|---|---|---|---|
| Faster RCNN | 640 × 480 | 143.5 | 41.14 | 56 |
| DeepLab v3+ | 640 × 480 | 119.8 | 39.76 | 58.8 |
| YOLOX pre-trained by Multi-Layer MoCo | 640 × 480 | 26.64 | 8.94 | 90.9 |

It can be seen from Table 5 that the number of parameters and the FLOPs of the YOLOX model pre-trained by the multi-layer MoCo method are far fewer than those of the Faster RCNN model and DeepLab V3+. The YOLOX model makes extensive use of the CSP module, so the size of the model can be scaled up or down by adjusting the number of channels. For the purpose of saving energy, reducing emissions, and limiting memory resources, we adopted the smallest model parameters, which are also convenient for deployment on edge devices with insufficient computing resources. The most intuitive benefits brought by the lower amount of computation are faster inference speed and higher FPS, which fully meets the industrial requirements for real-time damage detection of video streams shot inside aero-engines.

## 6. Conclusions

Based on the traditional MoCo method, we propose a multi-level contrastive learning based on the structural characteristics of the feature pyramid in the damage detection network. It is used in the pre-training target detection method to complete the damage detection task of aero-engines so that the encoder can learn the representation of samples at different scales, thereby optimizing and improving the representation ability of each layer of the damage detection network. We finally achieve the goal of efficiently detecting the internal damages of the aero-engine with low energy consumption, relieving the pressure of manual detection, and better guaranteeing the safe operation of the aero-engines. According to the experimental conclusion that the feature extraction ability of each layer of the detection network substantially increases by multi-layer contrastive learning, we reasonably believe that our proposed method can also have certain effects on other datasets.

**Author Contributions:** X.H. (Xing Huang) and L.L. conceptualized and designed the methodology and algorithm design for the study. P.D. has conducted research in related fields. D.Y. and J.Z. helped to modify the conception. X.H. (Xinjian Hu) and X.H. (Xing Huang) completed the experiments. L.L. led the writing of the manuscript. J.Z. reviewed the article's writing and helped to build datasets. All authors have read and agreed to the published version of the manuscript.

**Funding:** This research received no external funding.

**Acknowledgments:** This work was supported by the 173 program (2020-JCJQ-ZD-029).

**Conflicts of Interest:** The authors declare no conflict of interest.

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
