# Peer review of "An Aero-Engine Damage Detection Method with Low-Energy Consumption Based on Multi-Layer Contrastive Learning"

_electronics, doi:10.3390/electronics11132093_

Round 1

Reviewer 1 Report

In this manuscript, a multi-level contrastive learning method for aero-engine damage detection is proposed. The proposed method is a type of “encoder-decoder” structure, where the encoder is pre-trained where the decoder conducts the damage detection. The manuscript is well organised. The comments are:

1.       The title “” Green …” is a bit of misleading – how do you define and quantify your Green?

2.       When people talking about “real-time detection”, normally it is understood as applying the well-trained network as a model/predictor/classifier to an “instance” image. In the manuscript, the authors state “We use the aero-engine datasets to train the models and finally obtain three the computational speed, model size, and inference speed of each model.”, which indicates these are the training times (also refer to Table 5) – are you talking about online/ real-time training? Please explain why people need real-time training and it is important.

3.       In the experiments section, “The dataset used for pre-training includes 18,313 raw images captured in real scenes.” – It seems to me that all the results displayed in the section are the training results, where are the validation and testing results? If the trained model has very low generalisation capability, then it is useless. Please clarify your validation and test.

4.       There are some minor typos, say in the Abstract “transitional” should be “traditional”. Please check these throughout the manuscript.

Author Response

Dear reviewer:

Thank you very much for your attention and the referee’s evaluation and comments on our paper "Green Aero-Engine Damage Detection Based on Multi-Layer Contrastive Learning method". Those comments are all valuable and very helpful for revising and improving our paper, as well as the important guiding significance to our researches. We have studied comments carefully and have made correction which we hope met with approval. The main corrections in the paper and the responds to the reviewer's comments are as flowing:
Responds to the reviewer's comments:

1. Changing the title.

Reviewer: The title “ Green …” is a bit of misleading – how do you define and quantify your Green?

Reply: Thank you so much for pointing out. "Green" is literally a huge range, and this paper only proposes a surface damage detection method. Compared with other models, our proposed method has the advantage of low energy consumption, and has applicability in the industrial field of aero-engine damage detection, so the title of the paper has been modified, and it is changed to “An Aero-Engine Damage Detection method with Low-Energy consumption Based on Multi-Layer Contrastive Learning“.

2. 

Q: When people talking about “real-time detection”, normally it is understood as applying the well-trained network as a model/predictor/classifier to an “instance” image. In the manuscript, the authors state “We use the aero-engine datasets to train the models and finally obtain three the computational speed, model size, and inference speed of each model.”, which indicates these are the training times (also refer to Table 5) – are you talking about online/ real-time training? Please explain why people need real-time training and it is important.

Reply: Thank you so much for pointing our mistake out. We did not make it clear enough in the description of this paragraph of the original manuscript. We revised the description of experiment. In fact, the results shown in Table 6 are that we use several images to test and evaluate the performance of the trained model. In order to simplify the experiment, we directly used the 509 images in the training set to test the model computational performance,but it was wrongly stated in the original manuscript, which is unprofessional. Sorry for the confusion for reviewers. We have revised the correct representation in the new manuscript. The evaluation metrics we use are “FLOPS”, “number of parameters” and “FPS”. These evaluation metrics were not described or analyzed in the original manuscript. This is our shortcoming. Therefore in the new manuscript, we have added a description of these metrics in the Section5 Experiment, as well as an analysis of the experimental results. Table 6(original Table 5) is verified that our proposed method has great advantages over other baseline models in terms of computing power consumption and computing speed for the task of aero-engine damage image detection.

3. Adding a table to illustrate the training set and test set distributions in the experiment

Q: In the experiments section, “The dataset used for pre-training includes 18,313 raw images captured in real scenes.” – It seems to me that all the results displayed in the section are the training results, where are the validation and testing results? If the trained model has very low generalisation capability, then it is useless. Please clarify your validation and test.

A: Thank you for pointing out the inconsistency in this paper.  We made a mistake when writing the original manuscript. Instead of describing the training set and test set used for downstream tasks in the experiment, we only mentioned the 18,313 original images used for pre-training and the earlier fault annotations in section 4 in the original manuscript. The dataset obtained from the work. So, this is what creates confusion for the reader. We added table4 to the experimental section of section5, illustrating the datasets used for model tuning training and testing. Since more than ten thousand raw images are effective for contrastive learning, we consider our method suitable for this aero-engine damage detection task.

4. 

Q: There are some minor typos, say in the Abstract “transitional” should be “traditional”. Please check these throughout the manuscript.

A: Thank you so much for pointing out this problem in the manuscript. We have discussed the writing problems with a professional English supervisor, he gave us several suggestions, which have all been adopted and accordingly adjusted in the manuscript. We feel sorry for causing you unnecessary troubles in reviewing, we hope that the revised version may meet your exceptions. 

We sincerely hope this manuscript will be finally acceptable to be published on Electronics. Thank you very much for all your help and looking forward to hearing from you soon.

Best regards

Sincerely yours Lei Li

Reviewer 2 Report

Well written and supported with adequate data set. Important research for the sustainable future. 

Author Response

Dear reviewer:

Thank you so much for your decision and your approval comments on my manuscript. We have discussed the writing problems with a professional English supervisor, he gave us several suggestions, which have all been adopted and accordingly adjusted in the manuscript. We have tried our best to improve and made some changes in the manuscript.

We sincerely hope this manuscript will be finally acceptable to be published on Electronics. Thank you very much for all your help and looking forward to hearing from you soon.

Best regards

Sincerely yours Lei Li

Reviewer 3 Report

The paper introduces self-supervised contrastive learning for solving the problem of damage detection. The idea is interesting, however, some issues should be addressed:

First, it is not clear to me how the positive and negative samples are built. Do you use some hard negative sampling techniques to find more informative samples?

Second, the literature review is not through. Applications of contrastive learning in some relevant fields should be discussed, such as, Exploring cross-image pixel contrast for semantic segmentation and Regional Semantic Contrast, Aggregation for Weakly Supervised Semantic Segmentation and Rethinking Semantic Segmentation: A Prototype View. 

The datasets used in the experiments are very small-scale. I am think whether they are enough for contrastive learning, which basically needs large-scale samples to obtain promising performance.

Why is YoLoX selected for experiments? Is it possible to apply the proposed contrastive learning method to other detection models like Faster RCNN.

Author Response

Dear reviewer:

Thank you for your decision and constructive comments on my manuscript. We have carefully considered the suggestion of Reviewer and make some changes. We have studied comments carefully and have made correction which we hope met with approval.  We have tried our best to improve and made some changes in the manuscript.

According to the comments, the point-to-point revision instructions are as follows:

1.Adding one Figure and modifying the description of constructing the positive/negative samples.

Reviewer: First, it is not clear to me how the positive and negative samples are built. Do you use some hard negative sampling techniques to find more informative samples?

Reply: Thank you for pointing out. The original manuscript description was not clear enough. We add Figure 3 to the new manuscript to illustrate the construction of positive and negative samples. Taking Figure 3 as an example, after encoding the same picture, the above anchor sample encoder(encoder q) part shows the features q0, q1, q2 extracted by three layers of different depths, and the below Momentum CSPDarknet encoder (encoder k) part also shows features k0, k1, k2 extracted by the three layers of the same depth with YoLoX. q0 and k0 are from the same depth layer, q0 and k1, k2 are from different layers. So we set (q0, k0), (q1, k1), (q2, k2) as positive samples of each other, (q1, k0), (q2, k0), (q2, k1) are negative samples of each other. And in order to increase the number of negative samples, we set all the encoded features of different pictures at different levels as negative samples of each other. We didn't use any hard negative sampling techniques yet, but we got some inspiration and decided to add research in this idea in the future work.

2.Researching on applications of contrastive learning in some relevant fields and adding the literature review in Weakly Supervised Semantic Segmentation.

Reviewer: Second, the literature review is not through. Applications of contrastive learning in some relevant fields should be discussed, such as, Exploring cross-image pixel contrast for semantic segmentation and Regional Semantic Contrast, Aggregation for Weakly Supervised Semantic Segmentation and Rethinking Semantic Segmentation: A Prototype View.

Reply:  Thank you so much for the your suggestion. We did some research on applications of contrastive learning in some relevant fields. In subsection2.2, we added a reference to the paper of "Contrastive learning of Class-agnostic Activation Map for Weakly Supervised...".

3. Adding a table to illustrate the training set and test set distributions in the experiment

Reviewer: The datasets used in the experiments are very small-scale. I am think whether they are enough for contrastive learning, which basically needs large-scale samples to obtain promising performance.

Reply: Thank you for pointing out the inconsistency in this paper.  We made a mistake when writing the original manuscript. Instead of describing the training set and test set used for downstream tasks in the experiment, we only mentioned the 18,313 original images used for pre-training and the earlier fault annotations in section 4 in the original manuscript. The dataset obtained from the work. So, this is what creates confusion for the reader. We added table4 to the experimental section of section5, illustrating the datasets used for model tuning training and testing. Since more than ten thousand raw images are effective for contrastive learning, we consider our method suitable for this aero-engine damage detection task.

4. About the choice of YoLoX.

Reviewer: Why is YoLoX selected for experiments? Is it possible to apply the proposed contrastive learning method to other detection models like Faster RCNN.

Reply: Thank you for asking. We did a series of investigations. Since yolox is a one-step model, it is faster and consumes less energy than Faster RCNN. It can be seen from Table6 “Computational Speed, Model Size and Inference Speed Comparison”. We believe method can literally be applied to other models, but in this engineering projects which emphasize low carbon and environmental protection, we prior to consider real-time computing speed, computing energy consumption and other aspects, so we finally chose YoLoX.

We sincerely hope this manuscript will be finally acceptable to be published on Electronics. Thank you very much for all your help and looking forward to hearing from you soon.

Best regards

Sincerely yours Lei Li

Round 2

Reviewer 1 Report

The revised version is much better and can be accepted after English check.

Author Response

Reviewer: The revised version is much better and can be accepted after English check.

Reply:

  1. Some articles such as "to/ the" have been removed/added.
  2. Checked and fixed syntax errors: "integrate to(integrate into)";"stacked to(stacked into)"; "damages(damage)"; "remaining(remain)" and so on.
